# Identification of Phytogenic Compounds with Antioxidant Action That Protect Porcine Intestinal Epithelial Cells from Hydrogen Peroxide Induced Oxidative Damage

**DOI:** 10.3390/antiox11112134

**Published:** 2022-10-28

**Authors:** Jing Wang, Meixia Chen, Sixin Wang, Xu Chu, Haifeng Ji

**Affiliations:** 1Institute of Animal Husbandry and Veterinary Medicine, Beijing Academy of Agriculture and Forestry Sciences, Beijing 100097, China; 2Sino-US Joint Laboratory of Animal Science, Beijing Academy of Agriculture and Forestry Sciences, Beijing 100097, China; 3College of Agriculture and Animal Husbandry, Qinghai University, Xining 810016, China

**Keywords:** herbal plant, antioxidant, high throughput, intestinal epithelium, antibiotic alternatives

## Abstract

Oxidative stress contributes to intestinal dysfunction. Plant extracts can have antioxidant action; however, the specific phytogenic active ingredients and their potential mechanisms are not well known. We screened 845 phytogenic compounds using a porcine epithelial cell (IPEC-J2) oxidative stress model to identify oxidative-stress-alleviating compounds. Calycosin and deoxyshikonin were evaluated for their ability to alleviate H_2_O_2_-induced oxidative stress by measuring their effects on malondialdehyde (MDA) accumulation, reactive oxygen species (ROS) generation, apoptosis, mitochondrial membrane potential (MMP), and antioxidant defense. Nrf2 pathway activation and the effect of *Nrf2* knockdown on the antioxidative effects of hit compounds were investigated. Calycosin protected IPEC-J2 cells against H_2_O_2_-induced oxidative damage, likely by improving the cellular redox state and upregulating antioxidant defense via the Nrf2-Keap1 pathway. Deoxyshikonin alleviated the H_2_O_2_-induced decrease in cell viability, ROS production, and MMP reduction, but had no significant effect on MDA accumulation and apoptosis. *Nrf2* knockdown did not weaken the effect of deoxyshikonin in improving cell viability, but it weakened its effect in suppressing ROS production. These results indicate that the mechanisms of action of natural compounds differ. The newly identified phytogenic compounds can be developed as novel antioxidant agents to alleviate intestinal oxidative stress in animals.

## 1. Introduction

Oxidative stress refers to a redox imbalance caused by excessive free radical production in the body. It is an important contributor to animal diseases and reduces growth and production performance. When reactive oxygen species (ROS) and reactive nitrogen species (RNS) accumulate to excessively high levels, they cause irreversible damage to cell lipids, proteins, and DNA [1], thus affecting animal physiological function and production performance. In livestock and poultry production, numerous factors, including changes in the environment, physiological stages, and exogenous pathogenic toxins, can cause oxidative stress, thus destroying the redox balance in the body. Oxidative stress negatively affects animal health. Excess ROS in the body can damage the mucosal barrier of tissues and organs, resulting in functional damage and ultimately, disease [2,3]. Further, oxidative stress reduces production performance, reproductive performance, and animal product quality, thus having serious economic effects. In certain circumstances, such as at high temperatures, during weaning, and during pregnancy, supplementing exogenous antioxidants can effectively alleviate oxidative stress in animals, reduce oxidative damage, and improve health and production performance. With the development of the animal husbandry and feed industry in China, it is important to research and develop natural, green, and safe antioxidants to alleviate oxidative damage in livestock and poultry.

With the onset of the postantibiotic era, herbal plant additives are increasingly considered as nutritional strategies to sustain antibiotic-free production [4]. Herbal plant extracts and compounds have multiple biological properties, including antimicrobial, anti-inflammatory, antioxidant, and immunomodulatory actions, which improve intestinal health, growth performance, and production potential [5,6]. Numerous studies have investigated natural plant compounds and traditional Chinese medicine extracts for their antioxidant activities [7,8,9]. The use of many of them (e.g., carvacrol, hymol, catechins, quercetin, oregano, curcumin, and cinnamaldehyde) has been demonstrated in livestock animals as antioxidant agents for enhancing the antioxidant capacity of the body and relieving oxidative stress, thus improving their health and production performance [4,6,10]. These plant-derived antioxidants may exert antioxidant activity through scavenging excess oxygen free radicals and chelate metals [7,11] or enhancing the activity of antioxidant enzymes thus alleviating oxidative damage [12,13]. However, because of the complex composition of plant-derived products, the active ingredients and potential mechanisms are not fully unraveled, and in vitro platforms have been useful for identifying and understanding the mechanism of these potential plant-derived antioxidants [14,15].

To discover more effective oxidative-stress-alleviating natural compounds and identify the main ingredients with antioxidant action, the cell-based high-throughput screening (HTS) assays were developed for specific applications. Cardioprotective agents from a traditional Chinese medicine library have been identified using a high-throughput H_2_O_2_-induced oxidative-damage model in H9c2 cells [15]. In order to evaluate the intracellular oxidative stress in IPEC-J2 cells, a replicable and reliable medium-throughput setup based on CM-H2DCFDA fluorescence analysis was established by Ayuso et al. [16]. In the present study, using our established porcine epithelial cell (IPEC-J2) oxidative stress model, we screened a library of 845 traditional Chinese medicine extract monomers. The objective of this study was to discover new and potentially more potent plant-derived antioxidants. These newly identified plant antioxidants may have potential for further development as novel antioxidants for the prevention of oxidative stress in animals.

## 2. Materials and Methods

### 2.1. Chemicals and Traditional Chinese Herbal Compounds

A library of 845 pure, unique, natural compounds isolated from traditional Chinese herbal plants with known biological activities was purchased from Target Molecule (Shanghai, China). The collection includes more than 30 different types of chemicals including, but not limited to, flavonoids, phenols, alkaloids, coumarins, triterpenoids, phenylpropanoids, and monoterpenoids. The compounds were provided as 10-mM stocks in dimethylsulfoxide.

### 2.2. Cell Culture

The porcine intestinal epithelial cell line IPEC-J2 was kindly provided by Dr. Glenn Zhang (Oklahoma State University, Stillwater, OK, USA). The cells were cultured in DMEM/F12 complete medium (a 1:1 mixture of Dulbecco’s modified Eagle’s medium and Ham’s F-12; Gibco/Thermo Fisher Scientific, Waltham, MA, USA) containing 10% fetal bovine serum (Gibco), streptomycin (100 μg/mL), penicillin (100 U/mL), and 1% ITS premix (5 μg/mL insulin, 5 μg/mL transferrin, 5 ng/mL selenium) (ScienCell, San Diego, CA, USA) in a humidified incubator at 37 °C in the presence of 5% CO_2_. The cells were subcultured every 3–4 days.

### 2.3. Oxidative Stress Model Establishment

An in vitro oxidative stress model was established by treating IPEC-J2 cells with H_2_O_2_. IPEC-J2 cells were seeded in 96-well tissue culture plates (Costar, Corning Inc., Corning, NY, USA) at a density of 8000 cells/well in 200 μL of complete medium and incubated overnight plus 24 h. Then, the cells were treated with H_2_O_2_ at 0, 200, 400, 600, 800, 1000, 1200, or 1400 μM for 4 h. After the treatments, a cell counting kit (CCK)-8 (Dojindo, Kumamoto, Japan) was used according to the manufacturer’s instructions to detect cell viability. The absorbance at 450 nm was measured using a Multiskan FC instrument (Thermo Fisher Scientific, Waltham, MA, USA). Culture supernatants were collected and centrifuged at 1500 rpm for 10 min. Lactate dehydrogenase (LDH) levels in the supernatants were determined using an LDH assay kit (11644793001; Roche, Mannheim, Germany) and an automatic biochemical analyzer (Beckman Instruments, Brea, CA, USA).

### 2.4. Drug Toxicity Assay

Drug toxicity was evaluated before HTS. IPEC-J2 cells were seeded in 200 µL of medium in 96-well plates at 8000 cells/well and incubated overnight. The cells were treated with individual library compounds at 10 µM for 24 h. Cell viability was assessed using the CCK-8 assay to assess toxicity. The absorbance at 450 nm was measured using the Multiskan FC instrument.

### 2.5. HTS Assay

The HTS assay was conducted as previously described [15]. Briefly, IPEC-J2 cells were seeded in 96-well plates at 8000 cells/well. After overnight culture, the cells were stimulated with individual library compounds at 10 μM for 24 h. After incubation, the medium was discarded, and the cells were gently washed with warm phosphate-buffered saline (PBS). Then, H_2_O_2_ was added to each well at 1000 μM and the plate was incubated for an additional 4 h. Following H_2_O_2_ treatment, 10 µL of CCK-8 solution was added to each well. After 3 h of incubation, cell viability was determined by measuring the absorbance at 450 nm. For each compound, the cell viability after compound pretreatment plus H_2_O_2_ treatment was normalized to that after compound treatment alone. A compound for which cell viability that was higher than 70%, implying that cell viability was improved by 15–20% compared with that after H_2_O_2_ treatment, was considered a hit.

To assess the robustness of the HTS assay, the Z′-factor [17] was calculated. Cells in a 96-well plate were treated, and not treated, with 1000 µM H_2_O_2_ for 4 h and then subjected to the CCK-8 assay. The Z′-factor was calculated as follows:$$Z^{\prime }=1-\frac{\left(3\sigma_{p}+3\sigma_{n}\right)}{\left|{µ}_{p}-{µ}_{n}\right|}$$
where *σ_p_* and *σ_n_* are the standard deviations and *µ_p_* and *µ_n_* the means of positive and negative controls, respectively. Z′ ≥ 0.5 indicates that the assay can be used for HTS. The critical value was calculated to confirm the separation of the H_2_O_2_-induced cell damage model and untreated controls.

### 2.6. Secondary Screening and Validation of the Hit Compounds

Compounds for which the cell viability in the HTS assay was above 70% were further assayed at five concentrations (2.5, 5, 10, 20, and 40 μM) using the IPEC-J2 oxidative stress model in 96-well plates. Cells were grown overnight and then exposed to the hit compounds for 24 h. Then, cell viability was determined using the CCK-8 assay. Nontreated cells served as a control. The assay was performed three times.

### 2.7. Determination of Malondialdehyde (MDA) Accumulation and ROS Generation

IPEC-J2 cells were seeded in 6-well tissue culture plates at 2.5 × 10^5^ cells/well and incubated overnight. Then, the cells were pretreated with calycosin or deoxyshikonin at 2.5 or 5 μM for 24 h. After the incubation, the cells were exposed, and not exposed, to 1000 μM of H_2_O_2_ for 4 h. Nontreated cells served as a control. After the treatments, the cells were gently washed with PBS twice and lysed using RIPA lysis buffer (containing PMSF) (Solarbio, Beijing, China) for 10 min. After centrifugation at 10,000× *g*, 4 °C for 10 min, the supernatants were collected. Protein concentrations were determined using a bicinchoninic acid protein assay kit (Pierce, Madison, WI, USA). MDA concentrations were determined using a Porcine MDA ELISA kit (B162437; BIM Biosciences, San Francisco, CA, USA) according to the manufacturer’s instructions.

To assess ROS production, treated cells were digested with trypsin, washed twice with buffer solution, and incubated with ROS reagent (MAK143; Sigma-Aldrich, Madrid, Spain) for 30 min. ROS production was detected using a BD FACSCalibur flow cytometer (BD Biosciences, San Jose, CA, USA) and analyzed using the CellQuest software (BD Biosciences). The experiments were performed in triplicate.

### 2.8. Detection of Apoptosis and the Mitochondrial Membrane Potential (MMP)

After the treatments, apoptosis was detected by flow cytometry using an annexin V- phycoerythrin (PE)/7-aminoactinomycin D (7-AAD) assay kit (Solarbio). The cells were collected and resuspended in binding buffer. The cells were stained with 5 μL of annexin V/PE at room temperature in the dark for 5 min. Then, 10 μL of 20 μg/mL 7-AAD and 400 μL of PBS were added and apoptotic cells were immediately detected using the BD FACSCalibur flow cytometer and analyzed using the CellQuest software.

The MMP was evaluated using MitoScreen (JC-1) (551302; BD Biosciences, Franklin Lakes, NJ, USA), which uses the membrane-permeable dye JC-1 to detect mitochondrial depolarization in cells. After the treatments, the cells were digested with trypsin and washed twice with binding buffer. Then, JC-1 was added for 30 min after which the cells were suspended in binding buffer. MMP levels were detected by using the FACSCalibur flow cytometer and analyzed using the CellQuest software. MMP is reflected by the proportion of JC-1 aggregates and monomers. The experiments were performed in triplicate.

### 2.9. Determination of Antioxidant Activities

Total antioxidant capacity (T-AOC), catalase (CAT) activity, and heme oxygenase (HO-1) activity in cell lysates were measured using commercial kits to determine the antioxidant capacity of IPEC-J2 cells after the different treatments. T-AOC and CAT kits were purchased from the Nanjing Jiancheng Bioengineering Institute (Nanjing, China) and HO-1 ELISA kits were purchased from Enzo Life Sciences (Raamsdonksveer, The Netherlands). All assays were carried out following the manufacturer’s instructions. After the treatments, the cells were gently washed with PBS twice and then lysed using RIPA Lysis Buffer R2220 (containing PMSF) (Solarbio). Protein concentrations were determined as described method above. The proteins were stored at −20 °C until analysis. The experiments were performed in triplicate.

### 2.10. Total RNA Isolation and Quantitative Reverse Transcription (RT-q) PCR

After the treatments, cells were lysed using RNAzol RT (Molecular Research Center, Cincinnati, OH, USA) and total RNA was isolated by alcohol precipitation. RNA concentrations and quality were assessed using a NanoDrop spectrophotometer (Thermo Fisher Scientific). The RNA was transcribed into first-strand cDNA using an iScript cDNA synthesis kit (Bio-Rad, Hercules, CA, USA) according to the manufacturer’s instructions. qPCRs were run using iTaq Universal SYBR Green Supermix (Bio-Rad) in a QuantStudio 3 Real-Time PCR System (Thermo Fisher Scientific). Target genes were amplified using porcine gene primers (Appendix A) with 40 cycles of denaturation at 95 °C for 30 s, annealing at 60 °C for 30 s, and extension at 72 °C for 20 s. Target mRNA levels were normalized to that of the glyceraldehyde-3-phosphate dehydrogenase gene, whose expression level was not altered by any of the compounds applied. The 2^−ΔΔCt^ method was used to calculate relative fold changes in gene expression [18].

### 2.11. Western Blot Analysis

After the treatments, the cells were gently washed with PBS twice and total proteins were extracted and quantified as described above. Equal amounts of proteins were separated by sodium dodecyl sulfate gel electrophoresis and subsequently transferred to polyvinylidene difluoride membranes. After blocking with a PBST buffer containing 5% skim-milk at room temperature for 1 h, the blots were incubated with the indicated primary antibodies at 4 °C overnight. After three washes with tris-buffered saline, the blots were incubated with a horseradish peroxidase-conjugated secondary antibody for 1 h. The antibodies used in this study are listed in Appendix A. Immunoreactive bands were detected using an enhanced chemiluminescence kit (ECL-Plus; Thermo Fisher Scientific) and imaged using a Bio-Rad gel detection system.

### 2.12. Establishment of Nrf2-Knockdown IPEC-J2 Cells

Porcine *Nrf2* siRNA (sense: 5′-GCCCAUUGAUCUCUCUGAUTT-3′, antisense: 5′-AUCAGAGAGAUCAAUGGGCTT-3′) was synthesized at GenePharma (Shanghai, China). The siRNA was transfected into IPEC-J2 cells using Lipofectamine 3000 (Invitrogen, Carlsbad, CA, USA) as per the manufacturer’s instructions. Successfully transfected cells were treated with appropriate concentrations of calycosin or deoxyshikonin for 24 h. Then, the MDA concentration, ROS production, and apoptosis were detected as described above.

### 2.13. Statistical Analysis

Data were processed using GraphPad Prism version 6 (GraphPad Software, San Diego, CA, USA). All data are presented as mean ± standard error of the mean (SEM). Means of two groups were compared using an unpaired two-tailed Student’s *t*-test in SPSS (version 20.0, SPSS, Chicago, IL, USA). *p* < 0.05 was considered significant.

## 3. Results

### 3.1. Establishment of an Oxidative Stress Model for HTS

Oxidative agents markedly decrease cell viability. H_2_O_2_ is a commonly used strong oxidant. We developed an oxidative stress model for HTS of plant compounds by treating porcine IPEC-J2 cells with H_2_O_2_ at 0, 200, 400, 600, 800, 1000, 1200, or 1400 μM for 4 h and then assessed cell viability using the CCK-8 assay. As shown in Figure 1A, H_2_O_2_ concentration-dependently decreased the viability of IPEC-J2 cells. As 1000 μM H_2_O_2_ reduced cell viability to approximately 50% after 4 h, we used this concentration and treatment time in subsequent experiments. To further clarify the effect of H_2_O_2_ in inducing oxidative stress, we evaluated the effect of different concentrations of H_2_O_2_ on cell LDH leakage (Figure 1B). In agreement with the above findings, H_2_O_2_ concentration-dependently increased LDH leakage in IPEC-J2 cells, which indicated that H_2_O_2_ induces substantial cell membrane damage in these cells.

### 3.2. Variability and Robustness of the Model

To assess the quality of the oxidative stress model for use in cell-based HTS, cells were treated, and not treated, with 1000 μM H_2_O_2_ for 4 h. Cell viability was determined in 30 H_2_O_2_-treated wells and 30 nontreated wells and the data (optical density values) were used to analyze the variability between wells and the robustness of the oxidative stress cell model by calculating the Z’ factor. As shown in Figure 2, after treatment of IPEC-J2 cells with 1000 µM H_2_O_2_ for 4 h, the Z’ factor was 0.64, which is considered robust [18]. The critical values were 1.85% for the H_2_O_2_-free group and 1.48% for the H_2_O_2_-treated group. These results indicated that the H_2_O_2_-treated model group and the nontreated control group were well separated, and the assay could be used for HTS.

### 3.3. Identification of Active Monomers in a Chinese Herbal Compound Library

The established cell viability-based 96-well-plate HTS assay was used to identify compounds with antioxidant activity in a Chinese herbal compound library. Before the HTS, we screened out toxic compounds (inducing a >90% reduction in cell viability) using a drug toxicity assay. In total, 718 plant compounds were screened for their ability to alleviate the H_2_O_2_-induced reduction in cell viability. Compounds for which the cell viability was above 70% after H_2_O_2_ treatment were considered potential active compounds. The top 22 hits from this primary screening were selected for further validation (Figure 3A).

### 3.4. Secondary Screening of the Hit Compounds

A dose-response-based secondary HTS was conducted to validate the antioxidant activity of the 22 hits. Most compounds alleviated H_2_O_2_-induced oxidative damage at at least one of five concentrations examined (2.5, 5, 10, 20, and 40 μM) (Figure 3B). Different compounds had different optimal concentrations. Detailed information on these 22 compounds is provided in Table 1. Calycosin and deoxyshikonin were selected for further analysis of their antioxidant effect and their potential mechanisms. These two compounds have been shown to induce host defense peptides and to have immunomodulatory effects in porcine cells [19,20]. Further, they were effective at relatively low concentrations in the present study.

### 3.5. Characterization of the Antioxidant Activities of Calycosin and Deoxyshikonin

To further investigate the antioxidant activity of the two hit compounds, the lipid peroxidation product MDA, an indicator of oxidative damage, was first measured. The results demonstrated that H_2_O_2_ treatment significantly increased the MDA level in IPEC-J2 cells (*p* < 0.05; Figure 4A) compared with the control treatment. Calycosin and deoxyshikonin pretreatments decreased the MDA level compared with the H_2_O_2_ treatment alone; however, the effect of deoxyshikonin was not significant. Next, we evaluated intracellular ROS production. As shown in Figure 4B,C, ROS accumulation was significantly increased in IPEC-J2 cells after exposure to H_2_O_2_, whereas pretreatments with calycosin and deoxyshikonin significantly suppressed the ROS burst induced by H_2_O_2_. These results suggested that calycosin and deoxyshikonin have the potential to prevent cell lipid peroxidation and free radical accumulation, thus attenuating intestinal epithelial cell damage induced by H_2_O_2_.

### 3.6. Effects of Calycosin and Deoxyshikonin on the MMP

The MMP is affected by H_2_O_2_-induced ROS release. Therefore, we investigated the potential regulatory effects of calycosin and deoxyshikonin on the MMP during the process of relieving oxidative stress. H_2_O_2_-induced oxidative stress significantly decreased the MMP level in IPEC-J2 cells (Figure 5A,B). After calycosin and deoxyshikonin pretreatments, the MMP level was significantly increased compared with that after H_2_O_2_ treatment alone, demonstrating the positive effects of calycosin and deoxyshikonin on the mitochondrial redox state in H_2_O_2_-damaged cells.

### 3.7. Effects of Calycosin and Deoxyshikonin on the Apoptosis

Excess ROS induce apoptosis. Flow cytometry results demonstrated that H_2_O_2_ treatment significantly increased apoptosis. Calycosin and deoxyshikonin treatment and calycosin pretreatment significantly alleviated H_2_O_2_-induced apoptosis, whereas deoxyshikonin pretreatment had no effect (Figure 6A,B). To better understand the effects of calycosin and deoxyshikonin on apoptosis, the expression levels of the apoptosis-associated proteins Bcl-2, Bax, caspase 3, and cleaved caspase 3 in calycosin- or deoxyshikonin-pretreated H_2_O_2_-challenged IPEC-J2 cells were determined by western blotting (Figure 6C,D). H_2_O_2_ treatment lowered the expression of the antiapoptotic factor Bcl-2. Calycosin pretreatment significantly rescued Bcl-2 expression compared with H_2_O_2_ treatment, whereas deoxyshikonin pretreatment had no significant effect on Bcl-2 expression. H_2_O_2_ treatment enhanced the expression of the proapoptotic protein Bax. Calycosin and deoxyshikonin pretreatments significantly suppressed the increase in Bax expression. Finally, H_2_O_2_ treatment significantly increased the levels of caspase 3 and cleaved caspase 3, whereas calycosin and pretreatment significantly suppressed these increases and deoxyshikonin pretreatment slightly, albeit not significantly, lowered the increase of cleaved caspase 3.

### 3.8. Effects of Calycosin and Deoxyshikonin on the Antioxidant Defense System

The effects of calycosin and deoxyshikonin on the cellular antioxidant defense system were examined to further investigate the mechanisms of these two compounds in alleviating oxidative stress. As shown in Figure 7A, the mRNA levels of *HO-1*, *NQO1*, and *GPX1* were significantly elevated, indicating antioxidant defense system activation, in IPEC-J2 cells treated with H_2_O_2_. Compared with the levels after H_2_O_2_ treatment alone, calycosin and deoxyshikonin pretreatments alleviated *HO-1* and *NQO1* mRNA expression induced by H_2_O_2_, whereas they significantly increased *CAT* expression. The pretreatments had no significant effect on *GPX1* expression compared with H_2_O_2_ treatment alone. Calycosin treatment alone significantly upregulated all genes detected except *SOD1*, whereas deoxyshikonin treatment alone significantly increased the expression of *HO-1*, *NQO1*, and *CAT*.

The T-AOC was significantly increased in calycosin-treated cells, and pretreatments with calycosin and deoxyshikonin significantly attenuated the H_2_O_2_-induced decrease in T-AOC (Figure 7B). Next, HO-1 and CAT activities were evaluated (Figure 7C,D). In line with the gene expression results, after pretreatments with the two compounds, HO-1 activity was alleviated when compared with that after H_2_O_2_ treatment alone, whereas CAT activity was increased, although the effect of deoxyshikonin treatment was not significant. These results suggested that the effects of calycosin and deoxyshikonin may attributed to the activation of the endogenous antioxidant defense system.

### 3.9. Activation of the Nrf2 Signaling Pathway

To further explore the antioxidant mechanism of calycosin and deoxyshikonin, we evaluated (p-)Nrf2 expression in IPEC-J2 cells treated with the two compounds and H_2_O_2_. As shown in Figure 8A,B, H_2_O_2_ treatment significantly lowered p-Nrf2 expression, whereas calycosin pretreatment significantly reversed this reduction. However, pretreatment with deoxyshikonin had no significant effect on p-Nrf2 expression when compared with H_2_O_2_ treatment alone. Calycosin or deoxyshikonin treatment alone had no significant effect on p-Nrf2 expression compared to the control treatment.

### 3.10. Effects of Calycosin and Deoxyshikonin on IPEC-J2 Cells Treated with Nrf2 siRNA

To clarify the role of the transcription factor Nrf2 in oxidative stress alleviation by the two compounds, we established *Nrf2*-knockdown IPEC-J2 cells and evaluated cell viability, MDA production, ROS accumulation, and apoptosis after the treatments. *Nrf2* siRNA-transfected IPEC-J2 cells showed significantly lower cell viability than negative control cells when challenged with H_2_O_2_ (Figure 9A). The antioxidant effect of calycosin was blocked by *Nrf2* knockdown; knockdown cells showed significantly reduced viability when compared with negative control siRNA-transfected cells after pretreatment with calycosin and subsequent H_2_O_2_ treatment. The effect of deoxyshikonin pretreatment on cell viability was not significantly affected by *Nrf2* knockdown. Furthermore, *Nrf2* knockdown weakened the effects of calycosin in preventing MDA production and ROS generation in response to H_2_O_2_ challenge (Figure 9B–D). The effect of deoxyshikonin on ROS generation was also weakened after *Nrf2* knockdown (Figure 9C,D). *Nrf2* knockdown abolished the effect of calycosin in alleviating apoptosis, whereas deoxyshikonin pretreatment still had no effect on apoptosis after *Nrf2* knockdown (Figure 9E,F).

## 4. Discussion

We used a fast, in vitro HTS assay using IPEC-J2 cells to identify plant-derived compounds that have antioxidant capacity to provide a reference for later validation in animals and application in livestock production. In vitro assays have obvious advantages, including a low cost, short operation time, and the absence of ethical concerns [21,22,23]. Although differences exist between in vivo and in vitro cell responses to oxidative stress [24], in vitro assays allow for a preliminary selection.

The screening of 845 natural products led to the identification of 22 compounds that maintained cell viability above 70% after H_2_O_2_ treatment. These compounds were confirmed to alleviate the H_2_O_2_-induced reduction in cell viability and attenuated oxidative damage in a concentration-dependent manner. Phenols and flavonoids were dominant, accounting for approximately half of the total compounds identified. Studies have demonstrated that phenols and flavonoids have strong antioxidant potential [25,26]. Among the top 10 identified compounds, bisabolangelone, morroniside, calycosin, sinapine, forsythoside A, and baicalin have been reported to show antioxidant activity and mitigate oxidative damage caused by various diseases, mainly in humans, both in vitro and in vivo [27,28,29,30,31]. These compounds may exhibit anti-inflammatory, antibacterial, and other comprehensive effects along with their antioxidant activity.

Two hit compounds, calycosin and deoxyshikonin, were further validated as they were among the most potent compounds. Calycosin, originally extracted from the roots of *Astragalus membranaceus*, is a typical phytoestrogen with a wide range of pharmacological activities, including anticancer, anti-inflammatory, antioxidant, antiosteoporosis, neuroprotective, hepatoprotective, cardioprotective, antidiabetic, and proangiogenic and vasoprotective activities [32]. Wang et al. [33] evaluated the ABTS radical-scavenging ability and ferric ion-reducing antioxidant power of calycosin and confirmed its antioxidant activity. The oxidative-stress-alleviating effect of calycosin has been investigated using different cell lines and mouse models [34,35,36]. However, this study is the first to report the antioxidant effect of calycosin in porcine cells. Pretreatment with calycosin remarkably reduced ROS accumulation in IPEC-J2 cells stimulated with H_2_O_2_. Cellular ROS accumulation has been linked to several gastrointestinal tract disorders [37]. MDA is the main product of ROS-induced membrane lipid peroxidation, and oxidative damage induces intracellular MDA accumulation. Calycosin pretreatment significantly inhibited MDA production induced by H_2_O_2_. Previous studies have clearly demonstrated the inhibitory effects of pretreatment with calycosin on ROS and MDA accumulation in oxidative-stress-damaged cells or hosts, thus alleviating oxidative-stress-induced impairments [34,36,38,39].

Deoxyshikonin, extracted from *Lithosperraum erythrorhizon*, is a promising drug candidate for the treatment of wounds and cancers [40,41]. Few studies have directly demonstrated the antioxidant activity of deoxyshikonin. Park et al. [40] demonstrated that Lithospermi Radix extract had concentration-dependent free radical-scavenging activity. In the present study, deoxyshikonin pretreatment suppressed ROS accumulation and the decrease in cell viability, but not MDA levels, in IPEC-J2 cells treated with H_2_O_2_, which indicates its potential protective effect on oxidative-stress-induced damage in porcine epithelial cells. Shikonin, a deoxyshikonin analog, has been shown to inhibit H_2_O_2_-induced increases in ROS and MDA levels in human HT29 cells, indicating that it exhibits antioxidant activity and has potential for the treatment of oxidative-damage-associated diseases [42]. These and our findings warrant further exploration of the antioxidant potential of deoxyshikonin in humans and animals.

H_2_O_2_ markedly reduces cell viability by disrupting mitochondrial function and inducing apoptosis. H_2_O_2_ treatment decreases the MMP, thus disrupting mitochondrial function [34]. In H_2_O_2_-treated cells, proapoptotic gene and protein expression are upregulated and antiapoptotic gene and protein expression downregulated, resulting in apoptosis [43,44]. Accordingly, we found that H_2_O_2_ disrupted the MMP and induced apoptosis and apoptosis-related protein expression in IPEC-J2 cells. Calycosin pretreatment alleviated the reduction in MMP caused by H_2_O_2_ and prevented H_2_O_2_-induced apoptosis. Previous studies have suggested that calycosin increases the expression of Bcl-2 while reducing the expression of Bax and Bad, thus inhibiting apoptosis, in endothelial cells [45,46]. Other studies have reported that calycosin induces apoptosis, mainly in cancer cells. Calycosin triggered apoptosis of osteosarcoma 143B cells and MCF-7 cells via mitochondrial-dependent intrinsic apoptotic pathways [47,48]. In our study, calycosin alleviated H_2_O_2_-induced oxidative damage and apoptosis in porcine epithelial cells by ameliorating mitochondrial dysfunction and inhibiting mitochondrial-pathway-mediated apoptosis. Its mechanism of action requires further study. Deoxyshikonin pretreatment alleviated the disruptive effect of H_2_O_2_ on the MMP; however, it did not influence mitochondrial-mediated apoptosis. Likely, it may act on mitochondrial-mediated pyroptosis or other signaling pathways to alleviate H_2_O_2_-induced cell viability reduction, which requires further study. While shikonin has been demonstrated to attenuate H_2_O_2_-induced oxidative injury in HT29 human intestinal epithelial cells via antioxidant activities and the inhibition of mitochondrial-pathway-mediated apoptosis [42], the structural differences of the analogs may explain their differences in their activity and mechanism.

Nrf2 plays a key role in regulating oxidative-stress-derived damage [49]. In nonstressed conditions, Keap1, the main negative regulator of Nrf2, binds to Nrf2 to promote the degradation of Nrf2 in the cytoplasm [50]. Under oxidative stress and injury, Keap1 activity declines and Nrf2 is stabilized and translocates into the nucleus and activates the transcription of genes with antioxidant response elements. Numerous phytochemicals and natural bioactive agents activate the Nrf2-Keap1 pathway to increase the transcription of cytoprotective genes to protect cells against oxidative damage and combat certain diseases [51,52,53]. In the present study, the Nrf2-Keap1 pathway was significantly enhanced by calycosin pretreatment compared with H_2_O_2_ treatment. Meanwhile, the expression and activity of several antioxidant enzymes were upregulated, likely via the activated Nrf2 pathway. Calycosin has shown similar effects in other cell lines [29,54], suggesting that Nrf2 transcriptional activity is involved in the increased antioxidant enzyme expression and the cytoprotective effect of calycosin against H_2_O_2_-mediated oxidative injury. Deoxyshikonin pretreatment had no significant effect on p-Nrf2 expression following H_2_O_2_ challenge. Although several antioxidant enzymes were altered by deoxyshikonin treatment, we cannot conclude that the antioxidative effects of deoxyshikonin are mainly mediated by Nrf2-Keap1 pathway activation. It is highly likely that other signaling pathways are involved in its regulatory effects. We have previously found that deoxyshikonin strongly activated the MAPK signaling pathway and enhanced the gene expression of several antioxidant-related enzymes in IPEC-J2 cells based on RNA-sequencing obtained gene expression profiles after deoxyshikonin treatment [21], which may indicate that the MAPK pathway and other pathways may mediate the antioxidant and cytoprotective effects of deoxyshikonin.

To confirm the involvement of the Nrf2 signaling pathway, we evaluated MDA accumulation, ROS production, and apoptosis after *Nrf2* knockdown in IPEC-J2 cells. The cytoprotective effect of calycosin on H_2_O_2_-induced cell damage was abolished under Nrf2 deficiency. These results confirmed that the Nrf2 pathway is crucial for the cytoprotective effect of calycosin against H_2_O_2_-induced oxidative stress in porcine epithelial cells. Lu et al. [29] obtained similar results in brain astrocytes; they found that calycosin protected against H_2_O_2_-induced astrocyte oxidative injury via the AKT-Nrf2-HO-1 signaling pathway, and si*Nrf2* application dramatically abolished these effects. In rheumatoid arthritis synovial fibroblasts, calycosin significantly potentiated Nrf2 translocation by inducing p62 accumulation and Keap1 degradation to activate HO-1 and NQO1, thus suppressing IL-6 and IL-33 secretion and COX-2 mRNA and protein expression [54]. The anti-inflammatory effect of calycosin has been demonstrated in various in vivo and in vitro studies [35,55]. Whether calycosin exerts its cytoprotective effect in porcine epithelial cells via both antioxidant and anti-inflammatory actions is worthy of further investigation. *Nrf2* knockdown suppressed the effect of deoxyshikonin in reducing ROS accumulation but had no significant effect on the effects of deoxyshikonin on MDA and apoptosis levels, which further indicated that Nrf2 may not be the main signaling pathway, but is partially involved in the regulatory effect of deoxyshikonin on H_2_O_2_-mediated porcine epithelial cell damage. Shikonin attenuated oxidative stress in spiral ganglion neuron cells and spiral ganglion Schwann cells via Nrf2 pathway activation, thus improving neurological hearing damage in mice [56]. Despite their structural similarity, the analogs may have different mechanisms. More robust, in-depth studies are required to unravel the precise antioxidant mechanism of deoxyshikonin.

## 5. Conclusions

We successfully identified 22 natural products that alleviated H_2_O_2_-induced oxidative damage in IPEC-J2 cells using an HTS assay. Among them, calycosin and deoxyshikonin exhibited superior antioxidant capacity. Pretreatment with calycosin or deoxyshikonin alleviated H_2_O_2_-induced oxidative cell damage by regulating the redox state of cells and enhancing the antioxidant defense system. Nrf2-Keap1 signaling may be involved or partially involved in the protective mechanisms of the respective compounds. These oxidative-stress-alleviating natural compounds have the potential to be developed as novel therapeutic or protective agents for use in pigs and possibly, other livestock susceptible to oxidative stress. However, further research is required to fully understand the modes of action of the compounds. This was the first study to demonstrate the antioxidant effects of calycosin and deoxyshikonin in livestock animal cells. Further studies of the compounds in appropriate animal models will be needed to investigate their usability as antioxidants in livestock production.

## Figures and Tables

**Figure 1 antioxidants-11-02134-f001:**
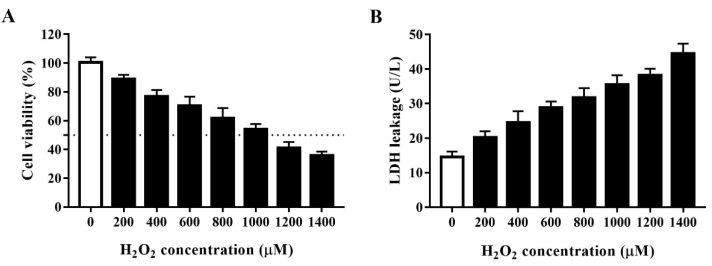
Establishment of an H_2_O_2_-induced oxidative stress model using IPEC-J2 cells. (**A**) Cell viability of IPEC-J2 cells after treatment with H_2_O_2_ at different concentrations. IPEC-J2 cells were incubated with the indicated concentrations of H_2_O_2_ for 4 h after which cell viability was measured using the cell counting kit (CCK)-8 assay. The black dotted line indicates 50% cell viability. (**B**) Lactate dehydrogenase (LDH) leakage from IPEC-J2 cells after treatment with H_2_O_2_ at different concentrations. Culture supernatants were collected after cell treatment and LDH leakage was determined using an LDH assay kit.

**Figure 2 antioxidants-11-02134-f002:**
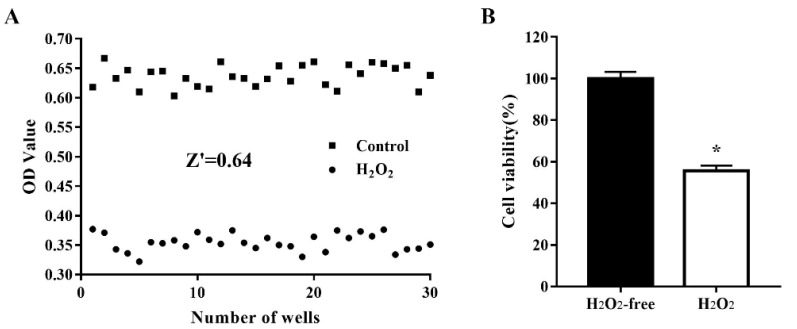
Reproducibility of the oxidative damage model in an automated 96-well assay format. IPEC-J2 cells were seeded in 96-well plates at a density of 8000 cells/well. Thirty H_2_O_2_-free wells were used as a positive control and 30 H_2_O_2_-treated wells were used as a negative control. The cells were treated, and not treated, with 1000 µM H_2_O_2_ at 37 °C for 4 h after which cell viability was assessed using the CCK-8 assay. The data are presented as optical density values of 30 individual wells (**A**) and as the mean ± standard error of the mean (SEM) of cell viability in 30 wells (**B**). * *p* < 0.05 vs. H_2_O_2_-free wells.

**Figure 3 antioxidants-11-02134-f003:**
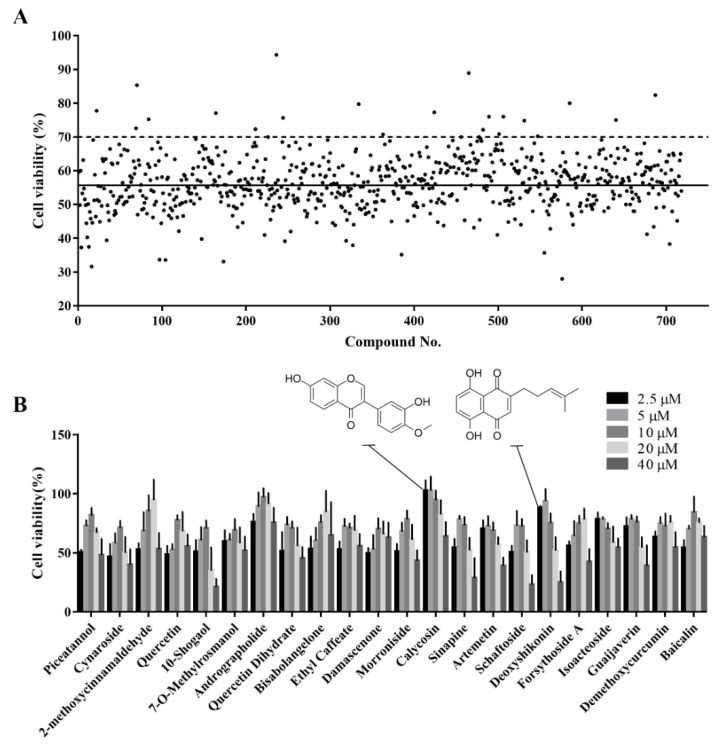
Identification of active monomers from the Chinese herbal compound library using a high-throughput screening assay. (**A**) Primary screening of 718 plant compounds. IPEC-J2 cells were pretreated with the compounds at 10 μM in 96-well plates for 24 h. Then, they were treated with 1000 μM H_2_O_2_ for 4 h after which cell viability was determined using the CCK-8 assay. Representative cell viability values for the 718 compounds are presented. The black solid line indicates the mean viability of cells treated with H_2_O_2_ alone and the black dotted line indicates 70% cell viability. Compounds that maintained cell viability above 70% were considered hits. (**B**) Secondary screening of the hit compounds at different concentrations. IPEC-J2 cells were pretreated with the hit compounds at 2.5, 5, 10, 20, or 40 μM in 96-well plates for 24 h. Then, they were treated with H_2_O_2_ at 1000 μM for 4 h after which cell viability was determined using CCK-8 assay.

**Figure 4 antioxidants-11-02134-f004:**
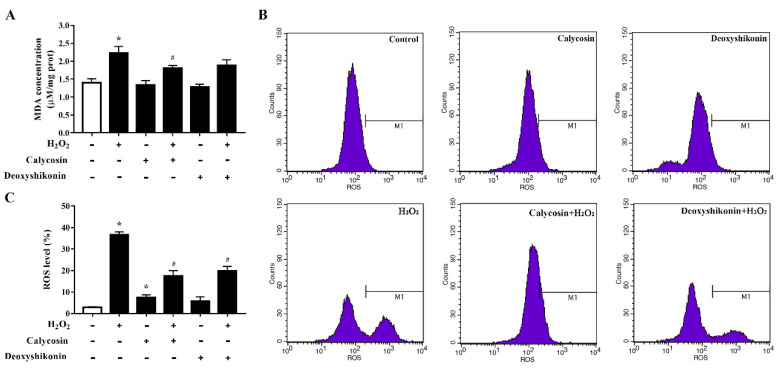
Effects of the hit compounds on H_2_O_2_-induced malondialdehyde (MDA) and reactive oxygen species (ROS) production in IPEC-J2 cells. IPEC-J2 cells were pretreated, and not pretreated, with calycosin or deoxyshikonin for 24 h, then with 1000 μM H_2_O_2_ for 4 h. Nontreated cells served as a control. (**A**) MDA concentrations were determined using an MDA assay. (**B**) Flow-cytometric analysis of ROS. (**C**) Quantification of ROS based on flow cytometry data. Data represent the mean ± SEM of three independent experiments. Differences between two groups were determined using Student’s *t*-tests. * *p* < 0.05 vs. nontreated control cells; ^#^ *p* < 0.05 vs. H_2_O_2_-treated cells.

**Figure 5 antioxidants-11-02134-f005:**
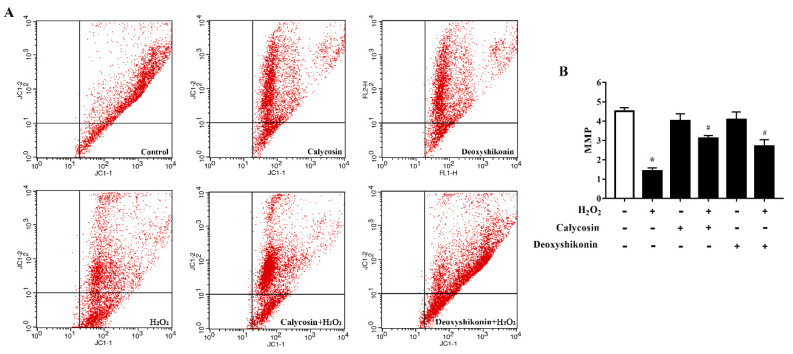
Effects of the hit compounds on H_2_O_2_-induced mitochondrial function reduction in IPEC-J2 cells. IPEC-J2 cells were pretreated, and not pretreated, with calycosin or deoxyshikonin for 24 h, then with 1000 μM H_2_O_2_ incubation for 4 h. Nontreated cells served as a control. (**A**) Flow-cytometric analysis of the mitochondrial membrane potential (MMP). (**B**) Quantification of the MMP based on flow cytometry data. Data represent the mean ± SEM of three independent experiments. Differences between two groups were determined using Student’s *t*-tests. * *p* < 0.05 vs. nontreated control cells; ^#^ *p* < 0.05 vs. H_2_O_2_-treated cells.

**Figure 6 antioxidants-11-02134-f006:**
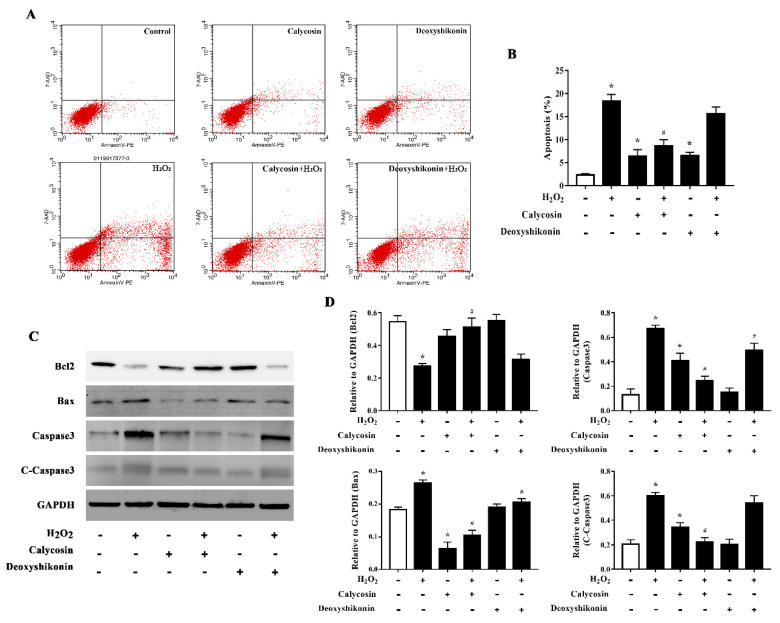
Effects of the hit compounds on H_2_O_2_-induced apoptosis in IPEC-J2 cells. IPEC-J2 cells were pretreated, and not pretreated, with calycosin or deoxyshikonin for 24 h, then with 1000 μM H_2_O_2_ for 4 h. Nontreated cells served as a control. (**A**) Flow-cytometric analysis of apoptotic cells. (**B**) Quantification of apoptotic cells based on flow cytometry data. (**C**) Western blots analysis of Bcl-2, Bax, caspase-3, and cleaved caspase-3 expression. (**D**) Quantitative analysis of apoptosis-related protein levels. Data represent the mean ± SEM of three independent experiments. Differences between two groups were determined using Student’s *t*-tests. * *p* < 0.05 vs. nontreated control cells; ^#^ *p* < 0.05 vs. H_2_O_2_-treated cells.

**Figure 7 antioxidants-11-02134-f007:**
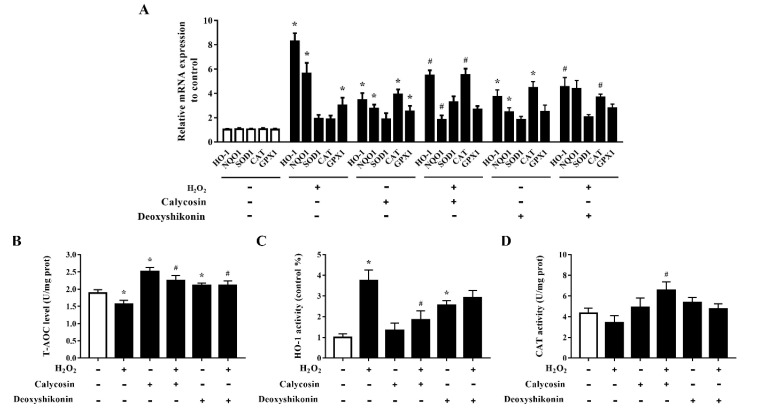
Effects of the hit compounds on antioxidant enzyme expression and activity in IPEC-J2 cells. IPEC-J2 cells were pretreated, and not pretreated, with calycosin or deoxyshikonin for 24 h, then with 1000 μM H_2_O_2_ incubation for 4 h. Nontreated cells served as a control. (**A**) mRNA levels of *HO-1*, *NQO1*, *SOD1*, *CAT*, and *GPX1* as determined by RT-qPCR. (**B**–**D**) T-AOC, HO-1, and CAT activities in lysed cells as determined using commercial assay kits. Data represent the mean ± SEM of three independent experiments. Differences between two groups were determined using Student’s *t*-tests. * *p* < 0.05 vs. nontreated control cells; ^#^ *p* < 0.05 vs. H_2_O_2_-treated cells.

**Figure 8 antioxidants-11-02134-f008:**
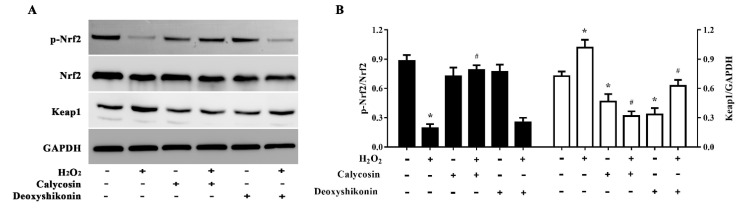
Effects of the hit compounds on the Nrf2-Keap1 pathway in IPEC-J2 cells. IPEC-J2 cells were pretreated, and not pretreated, with calycosin or deoxyshikonin for 24 h, then with 1000 μM H_2_O_2_ for 4 h. Nontreated cells served as a control. (**A**) Western blots of protein levels of Nrf2, p-Nrf2, and Keap1. (**B**) Quantitative analysis of Nrf2, p-Nrf2, and Keap1 protein levels. Data represent the mean ± SEM of three independent experiments. Differences between two groups were determined using Student’s *t*-tests. * *p* < 0.05 vs. nontreated control cells; ^#^ *p* < 0.05 vs. H_2_O_2_-treated cells.

**Figure 9 antioxidants-11-02134-f009:**
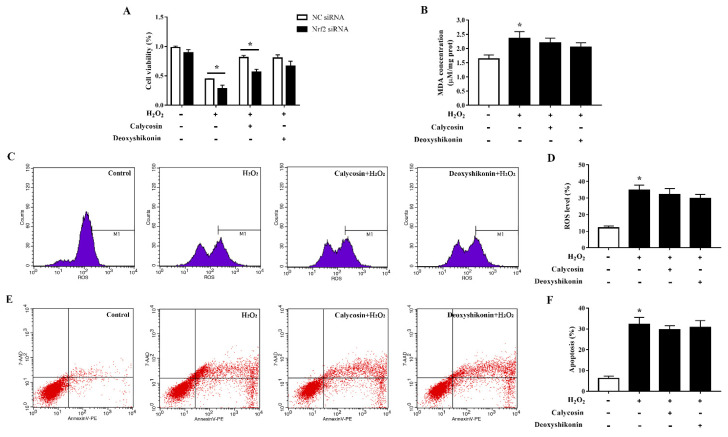
Effects of the hit compounds on H_2_O_2_-induced oxidative stress in Nrf2-knockdown IPEC-J2 cells. IPEC-J2 cells were transfected with Nrf2 siRNA or negative control siRNA and the efficiency of Nrf2 knockdown was confirmed by RT-qPCR. Nrf2-knockdown and control cells were treated, and not treated, with calycosin or deoxyshikonin for 24 h, then with 1000 μM H_2_O_2_ for 4 h. Nontreated cells served as a control. (**A**) Cell viability as measured using the CCK-8 assay. The results were normalized to negative control siRNA-transfected cells not treated with H_2_O_2_ and compounds. (**B**) MDA concentrations as determined using an MDA assay. (**C**) Flow-cytometric analysis of ROS. (**D**) Quantification of ROS using based on flow cytometry data. (**E**) Flow-cytometric analysis of apoptotic cells. (**F**) Quantification of apoptotic cells based on flow cytometry data. Data represent the mean ± SEM of three independent experiments. The means of two groups were compared using Student’s *t*-tests. * *p* < 0.05 vs. nontreated control cells.

**Table 1 antioxidants-11-02134-t001:** The cell viability and characteristics of the identified 22 hits at the indicated concentration.

Compounds	Cell Viability					CAS Number	Molecular Formula	Molecular Weight	Structural Family	Source
	2.5 μM	5 μM	10 μM	20 μM	40 μM					
Piceatannol	50.5	72.7	81.7	67.1	48.3	10083-24-6	C_14_H_12_O_4_	244.24	Alcohols	*Rheum officinale Baill.*
Cynaroside	46.7	58.2	71.2	50.0	40.1	5373-11-5	C_21_H_20_O_11_	448.38	Glycosides/Flavonoids	*Anthriscus sylvestris*
2-methoxycinnamaldehyde	52.8	68.4	85.4	94.6	53.2	1504-74-1	C_10_H_10_O_2_	162.18	Phenols	*Cinnamomum cassia*
Quercetin	48.9	52.6	77.8	68.4	55.5	117-39-5	C_15_H_10_O_7_	302.24	Glycosides/Flavonoids	*Fructus sophorae*
10-Shogaol	50.9	60.6	70.7	34.8	21.2	36752-54-2	C_21_H_32_O_3_	332.47	Phenols	*Zingiber officinale*
7-O-Methylrosmanol	59.6	60.7	69.1	58.3	51.8	113085-62-4	C_21_H_28_O_5_	360.45	Diterpenoids	*Rosmarinus officinalis*
Andrographolide	76.0	89.2	97.1	91.5	75.4	5508-58-7	C_20_H_30_O_5_	350.44	Terpenoids	*Andrographis paniculata*
Quercetin Dihydrate	51.5	73.7	70.8	55.8	45.3	6151-25-3	C_15_H_14_O_9_	338.27	Flavonoid	*Sophora japonica*
Bisabolangelone	53.3	60.2	75.8	84.7	64.9	30557-81-4	C_15_H_20_O_3_	248.32	Sesquiterpenoids	*Angelica sinensis*
Ethyl Caffeate	53.1	72.2	71.0	68.7	55.6	102-37-4	C_11_H_12_O_4_	208.21	Polyphenols	*Bidens pilosa*
Damascenone	49.4	53.0	70.1	65.8	63.1	23696-85-7	C_13_H_18_O	190.28	Ketone	*Rosa rugosa*
Morroniside	51.3	68.1	78.6	61.1	43.5	25406-64-8	C_17_H_26_O_11_	406.38	Iridoids	*Cinchona*
Calycosin	102.6	102.8	94.7	82.5	63.9	20575-57-9	C_16_H_12_O_5_	284.26	Flavonoids	*Astragalus membranaceus*
Sinapine	54.6	78.4	73.4	52.2	28.7	18696-26-9	C_16_H_24_NO_5_^+^	310.37	Alkaloids	*Raphanus sativus*
Artemetin	70.5	72.5	68.8	57.0	39.1	479-90-3	C_20_H_20_O_8_	388.37	Flavonoids	*Achillea millefolium*
Schaftoside	50.5	73.1	72.4	50.2	23.1	51938-32-0	C_26_H_28_O_14_	564.49	Flavonoids	*Desmodium styracifolium*
Deoxyshikonin	88.2	93.8	75.3	51.9	25.1	43043-74-9	C_16_H_16_O_4_	272.30	Quinones	*Lithosperraum erythrorhizon*
Forsythoside A	56.1	64.3	74.8	78.2	42.6	79916-77-1	C_29_H_36_O_15_	624.59	Phenylpropanoids	*Forsythia suspensa*
Isoacteoside	78.4	78.1	70.2	58.7	54.6	61303-13-7	C_29_H_36_O_15_	624.59	Phenylpropanoids	*Pedicularis striata Pall.*
Guaijaverin	72.5	78.5	76.0	54.6	39.1	22255-13-6	C_20_H_18_O_11_	434.35	Flavonoids	*Psidium guajava Linn.*
Demethoxycurcumin	63.4	74.8	72.6	75.8	54.5	22608-11-3	C_20_H_18_O_5_	338.36	Phenols	*Zingiber officinale Roscoe.*
Baicalin	54.3	69.9	84.3	75.6	63.3	21967-41-9	C_21_H_18_O_11_	446.36	Flavonoids	*Scutellaria baicalensis*

## Data Availability

The data are contained within the article and Appendix A.

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
