# Peer review of "Identification of Phytogenic Compounds with Antioxidant Action That Protect Porcine Intestinal Epithelial Cells from Hydrogen Peroxide Induced Oxidative Damage"

_antioxidants, 2022, doi:10.3390/antiox11112134_

Round 1

Reviewer 1 Report

In this study, the authors develop a high-throughput screen of >800 natural products to identify 22 natural products capable of reducing the oxidative damage caused to IPEC-J2 cells by hydrogen peroxide. The best two molecules (calycosin and deoxyshikonin) were selected for further study. The studies revealed that the Nrf2-Keap1 pathway is important for the action of calycosin, but less important for the action of deoxyshikonin. This suggests that alternative mechanisms may be operating in the case of deoxyshikonin. 

The manuscript is well presented in a logical manner, and presents some interesting results that will be of interest to biologists and biochemists working in the field of antioxidants.

I recommend this manuscript for publication in antioxidants after the following minor points have been addressed:

1. As the authors argue that calycosin and deoxyshikonin have different mechanisms of action, I believe that the reader would benefit from the inclusion of the chemical structures of calycosin and deoxyshikonin within the manuscript. This could possibly be included as an extension to figure 3 (i.e. Fig 3C).

2 Line 122: I believe that the subscripts are missing for p and n within the equation.

3. Line 137 (and also various other points in the manuscript): '...exposed or not to 1000 uM...' reads a little strange and might be better as '...exposed, and not exposed, to 1000 uM...'. Similarly, line 242 could become 'treated, and not treated, with 1000 uM...'

4. Some unnecessary capitalisation: Annexin (line 154), Tris-buffered (line 198). 

5. A space should be added between value and degrees centigrade (lines 140, 186, 198, 254 and maybe others).

6. Figure 9 caption: Check font size of lines 423425 (seems a bigger font size to the rest of the caption.

7. Reference list: Page number ranges should be longer en dash (–) instead of hyphen (-).

8. The volume numbers in the reference list should be italicised.

Author Response

Dear reviewer,

We appreciate your positive and constructive comments. We have considered them all and made changes as suggested in most cases. The following is our point-to-point response. Please note that all changes were highlighted in red in our revised manuscript, which we believe has been improved with incorporation of these changes. We hope that the revised manuscript is now suitable for publication.

Thank you for your consideration.

Sincerely,

Jing Wang

Comments:

  1. As the authors argue that calycosin and deoxyshikonin have different mechanisms of action, I believe that the reader would benefit from the inclusion of the chemical structures of calycosin and deoxyshikonin within the manuscript. This could possibly be included as an extension to figure 3 (i.e. Fig 3C).

RESPONSE: We agree with the comment. The chemical structures of calycosin and deoxyshikonin have been included in figure 3B.

  1. Line 122: I believe that the subscripts are missing for p and n within the equation.

RESPONSE: We apologized for this careless mistake. We have corrected the subscripts for and  within the equation.

  1. Line 137 (and also various other points in the manuscript): '...exposed or not to 1000 uM...' reads a little strange and might be better as '...exposed, and not exposed, to 1000 uM...'. Similarly, line 242 could become 'treated, and not treated, with 1000 uM...'

RESPONSE: We agree with the reviewer in that it is more accurate to use '...exposed, and not exposed, to 1000 uM...'. We have revised this kind of statements throughout the manuscript.

  1. Some unnecessary capitalisation: Annexin (line 154), Tris-buffered (line 198). 

RESPONSE: Changes have been made as suggested.

  1. A space should be added between value and degrees centigrade (lines 140, 186, 198, 254 and maybe others).

RESPONSE: The space between value and degrees centigrade has been added throughout the manuscript.

  1. Figure 9 caption: Check font size of lines 423–425 (seems a bigger font size to the rest of the caption.

RESPONSE: Thanks for reviewer’s comment, we have checked and revised font size.

  1. Reference list: Page number ranges should be longer en dash (–) instead of hyphen (-).

RESPONSE: Corresponding modification has been made according to the suggestion.

  1. The volume numbers in the reference list should be italicised.

RESPONSE: The volume numbers in the reference have been changed to italic.

Reviewer 2 Report

The rational behind the experiment was clear and straight forward. The manuscript is almost well written. 

There are some minor grammar issues that should be fixed in order to aid the accessibility of the results to the reader.

While many different sources are used to set up the study in the introduction, little previous evidence is stated. The introduction is thus short and poorly sets up the rationale for the study. More attention to how this study fits into previous work in oxidative stress and phytogenic compounds should be added to improve this section.

Please refer to doi: 10.3390/antiox9100992, 10.1016/j.phymed.2018.09.174, 10.1155/2017/4586068.

Author Response

Dear reviewer,

We appreciate your positive and constructive comments. We agree with your suggestion on our introduction. More information about previous work in oxidative stress and phytogenic compounds have been added. And we have carefully revised the introduction. Please note that all changes were highlighted in red in the revised manuscript, which we believe has been improved with incorporation of these changes. We hope that the revised manuscript is now suitable for publication.

Thank you for your consideration.

Sincerely,

Jing Wang

Reviewer 3 Report

In their manuscript Wang et al.  screened a large group of phytogenic compounds to identify oxidative stress-alleviating compounds using a porcine intestinal epithelial cell (IPEC-J2) oxidative stress model. The mechanisms investigated included their ability to alleviate H2O2-induced oxidative stress by measuring their effects on malondialdehyde (MDA) accumulation, reactive oxygen species (ROS) generation, apoptosis, mitochondrial membrane potential (MMP), antioxidant defense, and differ for the collection that included more than 30 different types of chemicals. The authors identified 22 natural products that alleviated H2O2-induced oxidative damage in IPEC-J2 cells using an HTS assay. Two hit compounds, calycosin and deoxyshikonin, were further validated as they were among the most potent compounds. The authors concluded that these oxidative stress-alleviating natural compounds have the potential to be developed as novel therapeutic or protective agents for use in pigs, but further studies of the compounds in appropriate animal models will be needed to investigate their usability as antioxidants in livestock production.

The manuscript is very clear, and written in an appropriate way, the question is original and well-defined, relevant for the field. The experimental model is well chosen, the methods are clearly presented and materials are sufficiently detailed that the research might be reproduced. Therefore, the manuscript’s results are reproducible based on the details given in the methods section, and the analyses are performed with the highest technical standards.

The results provide an advancement of the current knowledge and presented in a structured manner, the data are interpreted appropriately and consistently throughout the manuscript. The iconography (figures/ tables/ images/ schemes) are appropriate, easy to interpret and understand.

The statements and conclusions are consistent with the evidence and arguments presented, drawn coherent and supported by the listed citations.The cited references are mostly recent publications and relevant, and does not include an excessive number of self-citations.

Author Response

Dear reviewer,

We appreciate the many positive comments from you. We have revised the manuscript in detail according to the comments of other reviewers. We hope that the revised manuscript is now suitable for publication.

Thank you for your consideration.

Sincerely,

Jing Wang